# Astaxanthin Ameliorates Lipopolysaccharide-Induced Neuroinflammation, Oxidative Stress and Memory Dysfunction through Inactivation of the Signal Transducer and Activator of Transcription 3 Pathway

**DOI:** 10.3390/md17020123

**Published:** 2019-02-18

**Authors:** Ji Hye Han, Yong Sun Lee, Jun Hyung Im, Young Wan Ham, Hee Pom Lee, Sang Bae Han, Jin Tae Hong

**Affiliations:** 1Department of Pharmacy and Medical Research Center, Chungbuk National University, Osongsaengmyeong 1-ro, Osong-eup, Heungdeok-gu, Cheongju, Chungbuk 28160, Korea; aff4434@naver.com (J.H.H.); kallintz@gmail.com (Y.S.L.); junhyung091@gmail.com (J.H.I.); heepom@empas.com (H.P.L.); shan@chungbuk.ac.kr (S.B.H.); 2Department of Chemistry, Utah Valley University, 800 West University Parkway, Orem, UT 84058, USA; YHam@uvu.edu

**Keywords:** astaxanthin, lipopolysaccharide, Alzheimer’s disease, memory impairment, amyloidogenesis, neuroinflammation, oxidative stress, STAT3

## Abstract

Astaxanthin (AXT), a xanthophyll carotenoid compound, has potent antioxidant, anti-inflammatory and neuroprotective properties. Neuroinflammation and oxidative stress are significant in the pathogenesis and development of Alzheimer’s disease (AD). Here, we studied whether AXT could alleviate neuroinflammation, oxidative stress and memory loss in lipopolysaccharide (LPS) administered mice model. Additionally, we investigated the anti-oxidant activity and the anti-neuroinflammatory response of AXT in LPS-treated BV-2 microglial cells. The AXT administration ameliorated LPS-induced memory loss. This effect was associated with the reduction of LPS-induced expression of inflammatory proteins, as well as the production of reactive oxygen species (ROS), nitric oxide (NO), cytokines and chemokines both in vivo and in vitro. AXT also reduced LPS-induced β-secretase and Aβ_1–42_ generation through the down-regulation of amyloidogenic proteins both in vivo and in vitro. Furthermore, AXT suppressed the DNA binding activities of the signal transducer and activator of transcription 3 (STAT3). We found that AXT directly bound to the DNA- binding domain (DBD) and linker domain (LD) domains of STAT3 using docking studies. The oxidative stress and inflammatory responses were not downregulated in BV-2 cells transfected with DBD-null STAT3 and LD-null STAT3. These results indicated AXT inhibits LPS-induced oxidant activity, neuroinflammatory response and amyloidogenesis via the blocking of STAT3 activity through direct binding.

## 1. Introduction

Alzheimer’s disease (AD) is an age-related neurodegenerative disorder associated with increased production and aggregation of amyloid beta (Aβ), an insoluble peptide, which causes oxidative damage and neuroinflammation in the brain [1,2]. The pathological abnormalities in AD include a profound loss of synapses, microglial activation and increased memory dysfunction [3,4].

Several studies have demonstrated that lipopolysaccharides (LPS)—the endotoxins produced by gram-negative bacteria—induce neuropathological and behavioural changes in mice that are similar to those produced in a human AD brain [5,6,7]. LPS generate a chronic inflammatory and oxidative stress in brain that result in the production and accumulation of Aβs in both the cerebral cortex and hippocampus [8,9,10]. The reactive oxygen species (ROS) released under stress cause lipid peroxidation, leading to the formation of products, such as malondialdehyde (MDA) and downregulates the levels of anti-oxidants, such as glutathione (GSH), aiding the development of AD [2,9]. Cytokines and chemokines, such as tumour necrosis factor alpha (TNF-α), interleukin 1-beta (IL-1β), interleukin 6 (IL-6), monocyte chemoattractant protein 1 (MCP-1), macrophage inflammatory protein 1-alpha (MIP-1α) and macrophage inflammatory protein 1-beta (MIP-1β) exert inflammatory response that promote the loss of synapses and cognitive dysfunction [6,11]. The pro-inflammatory cytokines and oxygen radicals have been found abundantly in the brains of patients with AD [12,13]. In addition, neuroinflammation and oxidative stress produced by LPS results in the activation of microglia within the hippocampus and entorhinal cortex [14].

Signal transducer and activator of transcription 3 (STAT3), the members STAT family of proteins play critical roles in inflammatory diseases, including AD [15,16]. The post-mortem analysis of hippocampus retrieved from AD patients as well as the analysis of the brains of AD mouse models have shown the activation and elevated levels of STAT3 [17]. In several studies, the level of phosphorylated STAT3 was demonstrated to have increased in LPS-induced neuroinflammation in vivo [18] and in vitro [19]. In the pathogenesis of AD, the activation of STAT3 signalling pathway is positively associated with cytokine signalling during neuronal differentiation and inflammation [17,20]. We previously found that in transgenic mice overexpressing IL-32β, the phosphorylation of STAT3 was reduced, as were the levels of several cytokines, which reduced the memory impairment [21]. These data indicate that inhibition of STAT3 pathway and cytokine levels could be crucial for the reduction of memory impairment. The increased levels of ROS promote inflammatory processes and elevate the levels of pro-inflammatory cytokines [22]. Several studies have demonstrated that the phosphorylation of STAT3 is stimulated under oxidative stress [23,24,25]. In a previous study, we reported that STAT3 phosphorylation was upregulated by LPS-induced oxidative stress in astrocytes and microglial cells [24]. The depletion of GSH affected the phosphorylation of STAT3 in cardiac myocytes [25]. The oxidative stress-induced STAT3 phosphorylation could contribute to the development of several diseases. The blocking of STAT3 pathway protects against oxidative damages by increasing the levels of GSH [26]. Neuron-specific antioxidant oxidation resistance 1 delayed amyotrophic lateral sclerosis by delaying the activation of STAT3 [27]. These data indicate that oxidative stress could stimulate STAT3 and thus, accelerate the development of diseases. Moreover, STAT3 signalling was shown to trigger memory impairment and neuronal damage through the induction of neuroinflammation and oxidative stress [17,28].

Astaxanthin (AXT) is ubiquitous in nature, especially in the marine environment and is found in high amounts in the red-orange pigment of the shells of crustaceans (for example, crabs and, shrimp), salmon, trout and asteroidean [29]. It has been reported that AXT can protect skin from ultraviolet radiation-induced damages, ameliorate age-related macular degeneration, protect against chemically-induced cancers, increase high-density lipoproteins and enhance the immune system through its anti-oxidant and anti-inflammatory properties [29,30]. In previous studies, we have shown that anti-inflammatory and anti-oxidant agents prevent Aβ deposition by directly inhibiting the cleavage of amyloid precursor protein (APP) by γ-secretase [20,31,32,33], through the inhibition of STAT3 [34]. In the present study, we investigated whether AXT alleviates LPS-induced inflammatory response and memory impairment, both in vivo and in vitro.

## 2. Results

### 2.1. Astaxanthin Alleviates LPS-Induced Memory Impairment in Mice

Previous studies from our and other groups have demonstrated that memory impairment and amyloidogenesis can be induced by systemic injections of lipopolysaccharides (LPS) in in vivo and in vitro models [20,32,33,35]. To investigate the memory-improving effects of AXT using the LPS-induced memory impairment model, we performed water maze and passive avoidance tests. The ability of mice to learn and recall spatial memory through escape latency and distance was investigated in water maze two times per day for 6 days. The average escape latency and escape distance at the end of training were about 18.33 s and 242.83 cm in the control group. The LPS-injected mice exhibited an average escape latency and escape distance to the platform of about 35.44 s and 546.81 cm in day 6. However, the LPS-injected mice that were given AXT on day 6 showed a dose-dependent and significant decrease in escape latency to 28.7 and 22.94 s and in escape distance to platform to 396.02 and 310.31 cm in the 30 and 50 mg/kg administration groups (Figure 1A,B). After the final day of the water maze test, we performed a probe test to calculate the time spent in the target quadrant zone for testing the maintenance of memory function. The average time spent in the target quadrant was decreased in the LPS-injected mice (22.83%) compared to that in the control group mice (29.00%) but administration of AXT in the memory impaired mice increased the average time spent in the target quadrant to 27.54% at AXT 30 mg/kg administration group (Figure 1C). Our results indicated that the LPS-injected mice required more time to find the hidden platform and performed fewer platform crossing compared to the mice in the control group. However, the AXT-administered mice required less time to find the hidden platform and performed increased platform crossing compared to the LPS-injected mice. We then evaluated as to how long the mice could remember, through the passive avoidance test. Although there were no significant differences in the training trial, the step-through latency in the LPS-injected mice (29.67 s) decreased in comparison to that in the control group (131.39 s). However, the step-through latency in the LPS-injected mice recovered to 51.40 s in the 30 mg/kg AXT administration group and to 108.26 s in the 50 mg/kg AXT administration group in the testing trial (Figure 1D). Overall, our results demonstrated that LPS-induced memory impairment was alleviated by AXT.

### 2.2. Astaxanthin Downregulates LPS-Induced Aβ Burden in the Brain of Mice

To investigate the association between memory improvement and in the reduction of Aβ deposition as a result of AXT administration, we measured the Aβ level in the brain. The Aβ level in the brains of LPS-injected mice (152%) were higher than the levels in the control group but it was decreased in the brains of AXT-administered mice (Figure 2A). We also measured the activity of β-secretase in the brain, because Aβs are produced by activated β-secretases. The activity of β-secretase was increased in the brains of LPS-injected mice (123%) compared to that in the brains of the control group mice but it was decreased in the brains of AXT-administrated mice (Figure 2B). To confirm whether AXT could influence the inhibition of amyloidogenesis in the brain, we investigated the level of APP and β-secretase 1 (BACE1) proteins using western blot analysis. The expression levels of APP and BACE1 were observed to have increased in the brains of LPS-injected mice and the expression of APP was decreased in the 30 mg/kg AXT administration group and the expression of BACE1 was reduced by the administration of AXT (Figure 2C).

### 2.3. Astaxanthin Prevents LPS-Induced Neuroinflammation in the Brain of Mice

The activation of microglia is implicated in the neuroinflammation during the development of AD. To investigate the protective effect of AXT on the activation of astrocytes and microglia, we performed immunohistochemistry to detect the expression of glial fibrillary acidic protein (GFAP) (a marker protein of astrocytes), IBA-1 (a marker protein of microglia cells) and inflammatory proteins (iNOS and COX-2) in the CA3 and DG (dentate gyrus) regions of the brain of mice. The number of GFAP and IBA-1-reactive cells were lower in the AXT-administered mice compared to that in the LPS-injected mice, which was much higher than the number in the control group mice (Figure 3A). The number of iNOS and COX-2-reactive cells was also reduced in the AXT-administered mice compared to that in the LPS-injected (Figure 3B). The expression levels of GFAP, IBA-1, iNOS and COX-2 were further evaluated using western blot analysis. In consonance with the immunohistochemistry results, the increased expressions levels of these proteins by LPS were decreased in the AXT-administered mice (Figure 3C). However, the expression of GFAP was decreased at 30 mg/kg in the AXT-administered mice (Figure 3C). The production of pro-inflammatory cytokines is also involved in neuroinflammation and enhances the development of AD [36]. The pro-inflammatory cytokines also enhance the APP production and the process of proteolytic cleavage of APP to increase the production of Aβ [37]. To examine the production of a variety of pro-inflammatory cytokines and chemokines in the brain of mice, we performed real-time PCR. The levels of pro-inflammatory cytokines such as TNF-α, IL-1β and IL-6 and of chemokines, such as MCP-1, MIP-1α and MIP-1β were increased in the LPS-injected mice but their levels were decreased in the AXT-administered mice (Figure 4A,B).

### 2.4. Astaxanthin Reduces LPS-Induced Oxidative Stress in the Brain of Mice

Brain is particularly vulnerable to oxidative stress because of its high utilization of oxygen; therefore, oxidative stress has a crucial role in the pathogenesis of AD. Aβ may induce the production of ROS and ROS-related neurotoxic factors and thereby destroy the various types of biological molecules [9]. We determined the levels of oxidative stress in the brain of mice. Reduced glutathione (GSH) is a vital endogenous protective antioxidant against oxidative stress, which is oxidized to GSSG, referred to as the oxidized glutathione; the GSH/GSSG ratio and total GSH level are indices of the protective ability of cells under oxidative stress induced by toxicants. The GSH/GSSG ratio and total GSH level was observed to decrease in the LPS-injected mice compared to their values in the control group mice but it was increased by AXT treatment (Figure 4C). However, the level of thiobarbituric acid—a marker of lipid peroxidation—was elevated by LPS but was reduced by AXT treatment (Figure 4C).

### 2.5. Astaxanthin Inhibits Amyloidogenesis and Neuroinflammation in Microglia BV-2 Cells

Microglial cells are the primary LPS-responsive cells in the central nervous system (CNS). We performed western blot analysis, real-time PCR, β-secretase activity assay and nitro oxide and oxidative stress assays in BV-2 microglial cells to elucidate the neuroprotective, anti-inflammatory and anti-oxidant roles of AXT against LPS-induced neuroinflammation. The expression levels of APP and BACE1 protein were higher in response to LPS but these expression levels were inhibited by AXT treatment (Figure 5A). The LPS-induced β-secretase activity was decreased in the AXT-treated BV-2 cells at 20 μM (Figure 5B). We also detected the expression levels of iNOS, COX-2 and IBA-1 by western blot analysis. The expression levels of iNOS, COX-2 and IBA-1 were reduced in the AXT-treated BV-2 cells in a concentration-dependent manner (Figure 5C). We also determined the production of pro-inflammatory cytokines and chemokines in the BV-2 cells by real-time PCR. The levels of pro-inflammatory cytokines such as TNF-α, IL-1β and IL-6 and chemokines such as MCP-1, MIP-1α and MIP-1β were increased by LPS but were decreased in the AXT-treated BV-2 cells (Figure 6A,B).

### 2.6. Astaxanthin Inhibits LPS-Induced Oxidative Stress in the Microglia BV-2 Cells

Neuroinflammatory diseases, such as AD, are characterized by the reaction of superoxides with nitric oxide [38]. To determine whether AXT could reduce oxidative stress in microglia and could be used as a novel inhibitor, we determined the NO, total GSH and TBARS levels in the BV-2 cells. The concentration of NO was increased by LPS but it was decreased in the AXT-treated BV-2 cells (Figure 6C). The total GSH level was increased by LPS and was decreased by AXT treatment. However, there is no big difference between each group (Figure 6C). In addition, the level of TBARS was elevated by LPS but was decreased by AXT (Figure 6C).

### 2.7. Astaxanthin Inhibits the Phosphorylation of STAT3 in the Brain of Mice and the Microglia BV-2 Cells

To determine the involvement of the STAT3 pathway in the inhibitory effect of AXT, we investigated the interaction between AXT and STAT3. In the presence of AXT, the LPS-induced phosphorylation of STAT3 was inhibited in the LPS-injected mice at 50 mg/kg (Figure 7A). In addition, the LPS-induced STAT3 activation was reduced by AXT in the BV-2 cells (Figure 7B). The inhibition of phosphorylated STAT3 by STAT3 inhibitor was more effective than by AXT (Figure 7C). Moreover, the levels of the phosphorylated STAT3 showed a larger decrease when the BV-2 cells were co-treated with AXT and STAT3 inhibitor (Figure 7C).

### 2.8. Astaxanthin Directly Binds to the DBD and LD of STAT3

Astaxanthin showed inhibitory effects on STAT3 activation, both in vivo and in vitro. We investigated whether this could be associated with the physical interaction of AXT and STAT3. We found the potential binding mode of AXT to STAT3 and the precise conformation of AXT at the binding site in STAT3 via virtual docking analysis. Virtual docking analysis was performed using AutoDock VINA [39] and molecular graphics for the best binding model was generated using Discovery Studio Visualizer 2.0 (BIOVIA, San Diego, CA, USA). The docking model showed a surface rendering of the DNA-binding domain (DBD) (inside a binding pocket comprised of Ile258, Gln247, Ala250, Cys251, Glu324, Arg325 and Gln326) and linker domain (LD) (inside a binding pocket comprised of Asp334, Thr515, Lys573, Leu577, Ala578 and Lys574) of STAT3 with AXT; the binding affinity between the two was found to be -9.0 kcal/mol. To elucidate the interaction between AXT and the DBD and LD of STAT3, we performed a pull-down assay using AXT-conjugated epoxy-activated Sepharose 6B beads and cell lysate transfected with DBD-null STAT3 and LD-null STAT3 (Figure 8A). The AXT-conjugated beads pulled down the STAT3, whereas the vehicle control beads could not do so in the pull-down assay (Figure 8B). The luciferase activity of STAT3 was increased in the BV-2 cells transfected with wild-type STAT3, DBD-null STAT3 and LD-null STAT3 vectors by LPS. However, the AXT treatment decreased the STAT3 luciferase activity in the BV-2 cells transfected with wild-type STAT3 vector; however, the luciferase activity of STAT3 was not affected by the AXT treatment in the BV-2 cells transfected with the DBD-null STAT3 and LD-null STAT3 vectors (Figure 8C). In addition, the expression levels of APP and p-STAT3 proteins were downregulated in the BV-2 cells transfected with wild-type STAT3 vector; however, the expression levels of these protein were more decreased in the BV-2 cells transfected with wild-type STAT3 vector than in the BV-2 cells transfected with DBD-null STAT3 and LD-null STAT3 vectors and treated with AXT (Figure 8D). We also examined the production of pro-inflammatory cytokines (TNF-α and IL-6) and the ROS level (NO and TBARS) in the BV-2 cells transfected with wild-type STAT3, DBD-null STAT3 and LD-null STAT3 vectors. The levels of TNF-α and IL-6 were decreased by AXT in the BV-2 cells transfected with the wild-type STAT3 but these cytokine levels were not affected by the AXT treatment in the BV-2 cells transfected with the DBD-null STAT3 and LD-null STAT3 vectors (Figure 8E). The concentration of NO and TBARS were also reduced in the BV-2 cells transfected with the wild-type STAT3. However, the ROS levels were not inhibited by AXT in the BV-2 cells transfected with DBD-null STAT3 and LD-null STAT3 vectors (Figure 8F). Thus, AXT could contribute to the suppression of the STAT3 activity by binding to the DBD and LD domains of STAT3.

## 3. Discussion

Increasing evidence demonstrates that AXT possesses anti-oxidant and anti-inflammatory activities that could ameliorate the pathogenesis of AD [40,41]. In the present study, we showed that AXT could alleviate the LPS-induced neuroinflammation and oxidative stress resulting in the decrease in Aβ levels and in the activity of β-secretase in AXT-administered mice as well as in AXT-treated BV-2 microglia cells. The anti-oxidative, anti-inflammatory and anti- amyloidogenic effects were associated with the inhibitory effect of AXT on the memory impairment.

The pathogenesis of AD is critically associated with neuroinflammation in the brain [36]. Neuroinflammation occurs in the vulnerable regions of AD brain where there is high deposition of Aβ [42]. iNOS and COX-2 are known to be involved in inflammatory response [13,36]. iNOS is upregulated in the brain of patients with AD and iNOS knockout (KO) is protective in mouse models of AD [36,43]. COX-2 is also increased in the brain of AD patients [44,45]. In the brain of LPS-injected mice and LPS-treated BV-2 microglial cells, the immunoreactivity and expression of iNOS and COX-2 were increased compared to that in the control group; however, the expression of these proteins was decreased in the AXT-administered mice and AXT-treated BV-2 microglial cells. The astrocytes and microglial cells are activated and accelerate the progression of neurodegenerative diseases. In the brain of LPS-injected mice and LPS-treated BV-2 microglial cells, the immunoreactivity and expression of GFAP and IBA-1 were decreased in the AXT-administered mice and AXT-treated BV-2 microglial cells. The activation of astrocytes and microglial cells also releases a series of damaging cytokines and chemokines [46]. Cytokines contribute to nearly every aspect of neuroinflammation, including pro-inflammatory processes, chemoattraction and response of microglial cells to Aβ deposits [36]. Chemokines have been found to be upregulated in the brain of AD patients and recruit the astrocytes and microglial cells to the sites of Aβ deposition [47]. In the brain of LPS-injected mice and LPS-treated BV-2 microglial cells, TNF-α, IL-1β and IL-6 were also upregulated compared to their levels in the control group. However, these cytokines were downregulated, both in vivo and in vitro. MCP-1, MIP-1α and MIP-1β were also increased in the brain of LPS-injected mice and LPS-treated BV-2 cells compared to their levels in the control group; however, their levels were decreased in the AXT-administered mice and AXT-treated BV-2 microglial cells. These data indicate that the anti-inflammatory effects of AXT could contribute to the reduction in amyloidogenesis and thereby, memory impairment.

The CNS is particularly vulnerable to oxidative stress because of a high oxygen consumption rate, abundant unsaturated lipids and a relative deficit of antioxidant enzymes compared to other organs [9,48]. The oxidative stress may damage the various types of biological molecules, such as GSH and antioxidant enzymes [9]. Lipid peroxidation appears to be more marked in the AD patients. The end products of peroxidation, such as malondialdehyde and peroxynitrite, are also increased in the brain of patients with AD [49]. Moreover, oxidative stress can transform the non-aggregated Aβ into aggregated Aβ [50]. Aβ also stimulates ROS production in the microglia in rodents [51]. For example, lipid and protein peroxidation is increased in the APP/PS1 double knock-in mice [52]. The level of GSH was decreased in the APPsw/PS1dE9 double transgenic mice but the level of MDA was increased in the APPsw/PS1dE9 double transgenic mice [9]. In the present study, the GSH/GSSG ratio and the total GSH level were observed to recover and the TBARS level was decreased by AXT administration. Moreover, the NO level was also decreased in the AXT-treated microglial cells compared to its level in the LPS-treated BV-2 microglial cells. These data also indicate that the anti-oxidant property of AXT could be significant for reduction of amyloidogenesis and could, thus, reduce memory dysfunction induced by LPS.

STAT3 is abundantly expressed in brain and participates principally in the regulation of genes involved in inflammation [53]. In the neuronal compartment, the activation of STAT3 has been observed in senile plaques and it is involved in the neuronal loss by apoptosis in the brain [54]. LPS could upregulate neuroinflammation in microglia cells through the activation of STAT3 [55]. Pro-inflammatory cytokines affect the phosphorylation of STAT3 in transgenic AD mouse model [21,56]. Especially, IL-6 is known to activate STAT3, which is associated with memory impairment in the LPS-injected IL-6 KO mice [57]. Our results indicated that the expression of p-STAT3 was increased in LPS-injected mice and LPS-treated BV-2 microglia cells but these increases were prevented by AXT. These data indicate that the inhibitory effect of AXT on the STAT3 pathway could be significant for the anti-inflammatory and anti-oxidant effects of AXT and could, thus, be associated with the memory impairing effects of AXT.

Our results indicate that AXT plays an inhibitory role in LPS-induced neuroinflammation through the inactivation of STAT3. We investigated whether AXT could inhibit the STAT3 activity through binding of AXT and STAT3. Molecular docking simulations revealed that AXT directly binds to the DBD and LD domains of STAT3. As shown by luciferase and pull-down assays, the DBD-null STAT3 and LD-null STAT3 abolished the inhibitory effect of AXT on STAT3 activation. These data suggest that the DBD and LD are crucial for the AXT STAT3 interaction and imply that AXT may exert anti-inflammatory and anti-oxidative effects via the inhibition of STAT3 by directly binding to the DBD and LD domains. In conclusion, we demonstrate that AXT protects against LPS-induced mice AD model by inhibiting of the STAT3 activity, which could result in the inhibition of Aβ accumulation by attenuation of β-secretase activity through anti-oxidative and anti-neuroinflammatory properties. We, therefore, suggest that the formulation and synthetic modification of AXT could provide a potent therapeutic agent for the prevention of AD.

## 4. Materials and Methods

### 4.1. Animals Experiment and Housing Condition

Eight weeks old male imprinting control region (ICR) mice were purchased from DBL (Eumsung, Korea). The experiment was performed in accordance with the guidelines proscribed by the Chungbuk National University Animal Care Committee (CBNUA-929-16-01). The mice were acclimatized to the laboratory environment, maintained at 22 ± 1 °C and relative humidity of 55 ± 10%, with 12 h light-dark cycles throughout the experiment. All mice were fed a standard laboratory chow diet *ad libitum*. All mice were randomly divided into the following four groups (*n* = 8/group): control group, LPS group, LPS with AXT 30 mg/kg group, LPS with AXT 50 mg/kg group. The mice from AXT groups were daily administrated AXT that dissolved in olive oil for 4 weeks by oral gavage. Intraperitoneal (i.p.) injection of LPS (250 µg/kg) was administered to all groups except for the control group on the 4th week for 7 days. Control mice were given an equal volume of vehicle instead. Subsequently, the behavioural tests of learning and memory capacity were assessed using the water maze, probe and passive avoidance test after AXT/LPS administration. Mice were sacrificed after behavioural tests by CO_2_ asphyxiation.

### 4.2. Morris Water Maze

The water maze test is a widely accepted method for examining cognitive function and we performed this test as described by Morris et al. [58]. Maze testing was fulfilled by the SMART-CS (Panlab, Barcelona, Spain) program and equipment. A circular plastic pool (height: 35 cm, diameter: 100 cm) was filled with squid ink water kept at 22–25 °C. An escape platform (height: 14.5 cm, diameter: 4.5 cm) was submerged 1–1.5 cm below the surface of the water in position. The test was performed two times a day for 6 days during the acquisition phase, with two starting points of rotational starts. The position of the escape platform was kept constant. Each trial lasted for 60 s or ended as soon as the mouse reached the submerged platform. Escape latency and escape distance of each mouse were monitored by a camera above the centre of the pool connected to a SMART-LD program (Panlab, Barcelona, Spain). A quiet environment, consistent lighting, constant water temperature and a fixed spatial frame were maintained throughout the experimental period.

### 4.3. Probe Test

To assess memory consolidation, a probe test was performed 24 h after the water maze test. For the probe test, the platform was removed from the pool which was used in the water maze test and the mouse were allowed to swim freely. The swimming pattern of each mouse was monitored and recorded for 60 s using the SMART-LD program (Panlab, Barcelona, Spain). Consolidated spatial memory was estimated by the time spent in the target quadrant area.

### 4.4. Passive Avoidance Test

The passive avoidance response was determined using a “step-through” apparatus (Med Associates Inc., Fairfax, VT, USA) that is divided into an illuminated compartment and a dark compartment (each 20.3 × 15.9 × 21.3 cm) adjoining each other through a small gate with a grid floor, 3.175 mm stainless steel rods set 8 mm apart. 48 h after the probe test, a training trial was performed. The mice were placed in the illuminated compartment facing away from the dark compartment for the training trial. When the mice moved completely into the dark compartment, it received an electric shock (0.45 mA, 3 s duration). Then, the mice were returned to their cage. 24 h after the training trial, each mouse was placed in the illuminated compartment and the latency period to enter the dark compartment defined as “retention” was measured. The time when the mice entered into the dark compartment was recorded and described as step-through latency. The retention trials were set at a cut-off time limit of 180 s.

### 4.5. Collection and Preservation of Brain Tissue

After the behavioural tests, mice were anaesthetized with CO_2_ gas and then perfused with phosphate-buffered saline (PBS). The brains were immediately removed from the skull and the tissues were divided in half. One stored at −80 °C, the other was fixed in 4% paraformaldehyde for 72 h at 4 °C. The brains were transferred to 30% sucrose solutions, respectively.

### 4.6. Microglial BV-2 Cells Cultures

Microglial BV-2 cells were obtained from the American Type Culture Collection (Rockville, MD, USA). Microglial BV-2 cells were maintained with serum-supplemented culture media of DMEM supplemented with FBS (10%) and antibiotics (100 units/mL). The microglial BV-2 were incubated in the culture medium in a humidified incubator at 37 °C and 5% CO_2_. The cultured cells were treated with several concentrations (5, 10, 20 μM) of AXT, 3 h before LPS (1 µg/mL) addition. The cells were harvested after 24 h.

### 4.7. Western Blot Analysis

Homogenized brain tissues were lysed by protein extraction solution (PRO-PREP, iNtRON, Sungnam, Korea) and the total protein concentration was determined using the Bradford reagent (Bio-Rad, Hercules, CA, USA). 40 µg of extracted protein were separated by SDS/PAGE and transferred to Immobilon^Ⓡ^ PVDF membranes (Millipore, Bedford, MA, USA). The membrane was blocked with 5% dried skimmed milk for 1 h at room temperature, followed by incubation with specific primary antibodies for overnight at 4 °C. The membranes were washed with Tris-buffered saline containing 0.05% Tween-20 (TBST) and incubated with diluted horse radish peroxidase-conjugated secondary antibodies for 1 h at room temperature. After washes, the binding of antibodies to the PVDF membrane was detected using the Immobilon Western Chemilum HRP substrate (Millipore, Bedford, MA, USA). The band intensities were measured using the Fusion FX 7 image acquisition system (Vilber Lourmat, Eberhardzell, Germany) and quantified using Image J software. Specific primary antibodies were purchased from Santa Cruz Biotechnology (GFAP, p-STAT3, STAT3 and β-actin; Dallas, TX, USA), Cell signalling Technology (iNOS and COX-2; Trask Lane, Danvers, MA, USA) and Abcam (APP, BACE1 and IBA-1; Cambridge, MA, USA). Secondary antibodies were purchased from Santa Cruz Biotechnology (anti-mouse, anti-rabbit and anti-goat; Dallas, TX, USA).

### 4.8. Immunohistochemistry

After being transferred to 30% sucrose solutions, brains were cut into 20 µm sections by using a cryostat microtome (Leica Microsystems, Seoul, Korea). After two washes in PBS (pH 7.4) for 10 min each, endogenous peroxidase activity was quenched by incubating the samples in 3% hydrogen peroxide in PBS for 20 min. The sections were blocked for 1 h in 5% bovine serum albumin (BSA) solution and incubated with specific primary antibodies for overnight at 4 °C. And then, the sections were washed twice for 10 min each in PBS and incubated in biotinylated anti-mouse or anti-rabbit or anti-goat IgG-horseradish peroxidase (HRP) secondary antibodies for 90 min. The sections were washed three times for 10 min each in PBS and visualized by a chromogen DAB (Vector Laboratories) reaction for up to 10 min. Finally, the sections were dehydrated in ethanol, cleared in xylene, mounted with Permount (Fisher Scientific, Hampton, NH) and evaluated on a light microscope (Nikon, Tokyo, Japan) and the quantitated positive cells manually. We investigated the region of brain from CA3 and DG region in hippocampus for anatomical studies. CA3 and DG network is the most critical for contributing to memory storage and retrieval of memory sequences. Specific primary antibodies were purchased from Santa Cruz Biotechnology (GFAP; Dallas, TX, USA) Cell signalling Technology (iNOS and COX-2; Trask Lane, Danvers, MA, USA) and Abcam (IBA-1; Cambridge, MA, USA).

### 4.9. Measurement of Aβ_1–42_

Lysates of brain tissue were obtained through protein extraction buffer containing protease inhibitor and were centrifuged at 14,000 rpm for 30 min to extract protein. Aβ_1–42_ levels were determined using each specific mouse amyloid-beta peptide 1-42 ELISA Kit (CUSABIO, Carlsbad, CA, USA). In brief, 100 μL of sample was added into a precoated plate and incubated for 2 h at 37 °C. After removing any unbound substances, a biotin-conjugated antibody specific for Aβ_1–42_ was added to the wells. After washing, avidin-conjugated HRP was added to the wells. Following a wash to remove any unbound avidin-enzyme reagent, a substrate solution was added to the wells and the colour was developed in proportion to the amount of Aβ_1–42_ bound in the initial step. The the colour was development was stopped and the intensity of the colour was measured.

### 4.10. Assay of β-Secretase Activities

β-secretase activity in the mice brains was determined using a commercially available β-secretase activity kit (Abcam, Cambridge, MA, USA). Solubilized membranes were extracted from brain tissues using β-secretase extraction buffer, incubated on ice for 1 h and centrifuged at 5000 × g for 10 min at 4 °C. The supernatant was collected. A total of 50 μL of sample (total protein 100 µg) or blank (β-secretase extraction buffer 50 μL) was added to each well (used 96-well plate) followed by 50 μL of 2  ×  reaction buffer and 2 μL of β-secretase substrate incubated in the dark at 37 °C for 1 h. Fluorescence was read at excitation and emission wavelengths of 335 and 495 nm, respectively, using a fluorescence spectrometer (Gemini EM, Molecular Devices, CA, USA).

### 4.11. RNA Isolation and Quantitative Real-Time RT-PCR

Total RNA from brain tissues were extracted by RiboEx^TM^ Total RNA isolation solution (GeneAll Biotechnology, Seoul, Korea) and cDNA was synthesized using a High Capacity RNA-to-cDNA kit (Applied Biosystems, Foster City, CA, USA). Quantitative real-time RT-PCR was performed on a 7500 real-time PCR system (Applied Biosystems, Foster City, CA, USA) for custom-designed primers and β-actin was used for house-keeping control using a QuantiNova SYBR Green PCR kit (Qiagen, Hilden, Germany). Cycling conditions consisted of a denaturation of 5 s at 95 °C and a combined annealing/extension of 10 s at 60 °C followed by 40 cycles. The values obtained for the target gene expression were normalized to β-actin and quantified relative to the expression in control samples.

### 4.12. Nitro Oxide and Oxidative Stress Assay

Nitro oxides (NO) were measured according to the manufacturer’s protocol (iNtRON, Sungnam, Korea). Hydrogen peroxides assay was performed as described in the manufacturer’s protocol (Cell Biolabs, San Diego, CA, USA). Reduced glutathione (GSH) and oxidized glutathione (GSSG) were measured using GSH/GSSG Ratio Detection Assay Kit (Abcam, Cambridge, MA, USA). Lipid peroxidation was measured by determining the generation of malondialdehyde (MDA; TBARS Assay kit, Cayman, Ann Arbor, MI, USA).

### 4.13. Plasmid Construction

The coding region of Mus musculus STAT3 was amplified by PCR using full-length M. musculus STAT3 cDNA as a template. Purified PCR products were double-digested with EcoRI and XhoI and then sub-cloned into the pcDNA3.1 vector containing a cytomegalovirus promoter, pUC origin and an ampicillin- resistance gene. STAT3 deletion mutant (DBD deletion, LD deletion) was generated by Bionics (Seoul, Korea) and the mutants were checked by sequencing.

### 4.14. Pull-Down Assay

AXT (1 mg) was dissolved in 1 mL of coupling buffer (0.1 M NaHCO3, pH 11.0 containing 0.5 M NaCl) and conjugated with epoxy-activated Sepharose 6B (GE Healthcare Korea, Seoul, Korea). The epoxy-activated Sepharose 6B was swelled and washed in distilled water on a sintered-glass filter and then washed with the coupling buffer. The epoxy-activated Sepharose 6B beads were added to the AXT-containing coupling buffer and rotated at 4 °C overnight. After washing, unoccupied binding sites were blocked with 0.1 M Tris-HCl (pH 8.0) for 2 h at room temperature. The AXT-conjugated Sepharose 6B was washed with three cycles of alternating pH wash buffers (buffer 1: 0.1 M acetate and 0.5 M NaCl, pH 4.0; buffer 2: 0.1 M Tris-HCl and 0.5 M NaCl, pH 8.0). The control unconjugated epoxy-activated Sepharose 6B beads were prepared as described above in the absence of AXT. The cell lysate was mixed with AXT-conjugated Sepharose 6B or with Sepharose 4B at 4 °C overnight. The beads were then washed three times with TBST. The bound proteins were eluted with SDS loading buffer. The proteins were resolved by SDS-PAGE followed by immunoblotting with an antibody against STAT3 (1:1000 dilutions, Santa Cruz Biotechnology).

### 4.15. Statistical Analysis

The data were analysed using the GraphPad Prism 4 version 4.03 software (Graph-Pad Software, La Jolla, CA, USA). Data are presented as mean ± SD. When the P value in the analysis of variance test indicated statistical significance, the differences were assessed by the Tukey’s test. A value of *p* ≤ 0.05 was considered to be statistically significant.

## Figures and Tables

**Figure 1 marinedrugs-17-00123-f001:**
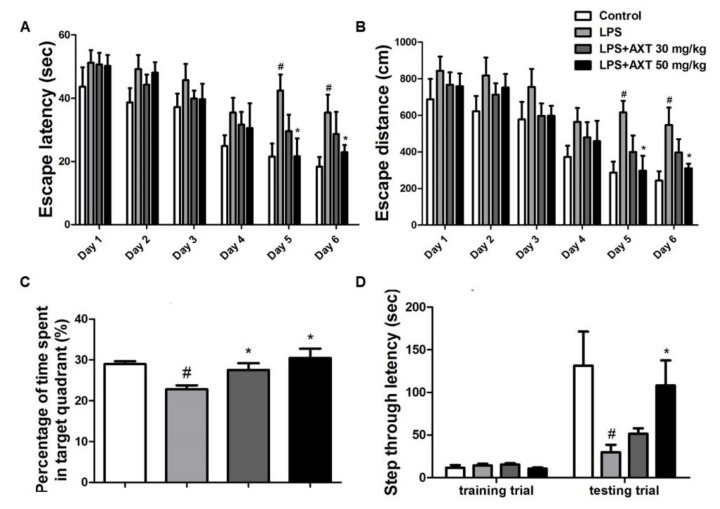
Effect of astaxanthin (AXT) on lipopolysaccharide (LPS)-induced improvement of memory impairment in the brain of mice. The mice (*n* = 8) were daily administrated AXT by oral gavage at dose of 30 or 50 mg/kg for 4 weeks. I.p. injection of LPS (250 μg/kg) was administrated except for control group on the 4th week for 7 days and they were evaluated for learning and memory of spatial information using the water maze. (**A**) Escape latency, the time required to find the platform and (**B**) escape distance, the distance swam to find the platform were measured. After the water maze test, (**C**) probe test to measure maintenance of memory were performed. The time spent in the target quadrant and target site crossing within 60 s was represented. (**D**) A passive avoidance test was performed by step-through method. *n* = 8 per group. The data are shown as the means ± SD of the mean. ^#^
*p* < 0.05 control group vs. LPS group, * *p* < 0.05 LPS-group vs. LPS with AXT group.

**Figure 2 marinedrugs-17-00123-f002:**
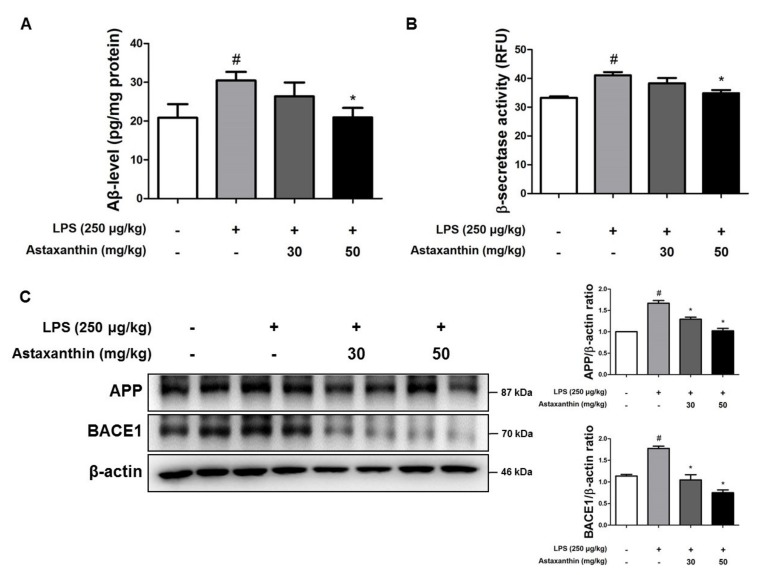
Effect of astaxanthin on LPS-induced Aβ accumulation and expression of amyloidogenic protein in the brain of mice. (**A**) The levels of Aβ_1-42_ in the brain of mice were assessed using a specific Aβ ELISA. *n* = 4 per group (**B**) The β-secretase activity in the brain of mice was measured using assay kit. *n* = 4 per group (**C**) The expression of APP and BACE1 were detected by western blot using specific antibodies in the brain of mice. β-actin protein was used as an internal control and graphs represented the arbitrary density of blot signal. *n* = 4 per group. The data are shown as the means ± SD of the mean. ^#^
*p* < 0.05 control group vs. LPS group, * *p* < 0.05 LPS-group vs. LPS with AXT group.

**Figure 3 marinedrugs-17-00123-f003:**
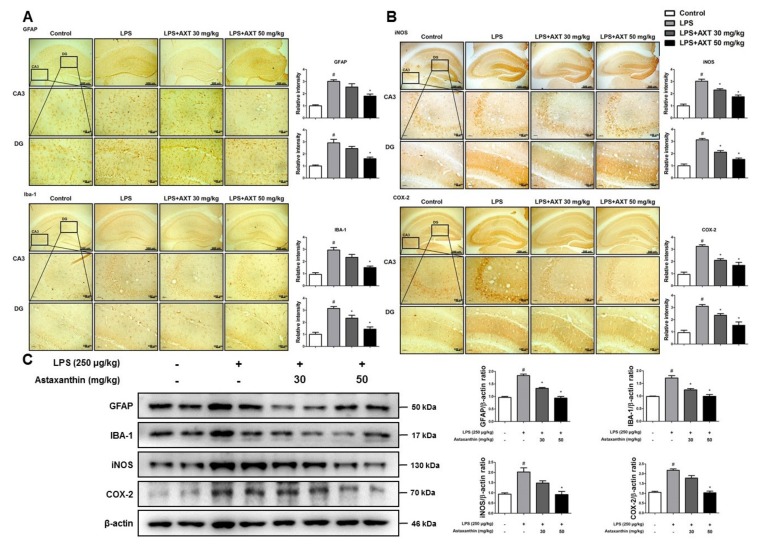
Effect of astaxanthin on LPS-induced neuroinflammation and amyloidogenesis in the brain of mice. Immunohistochemical analysis of (**A**) glial fibrillary acidic protein (GFAP), IBA-1 (**B**) iNOS and COX-2 antibodies were investigated two different regions (CA3; cornu ammonis 3 and DG; dentate gyrus) in 20-µm-thick sections of the brain hippocampus of mice with specific primary antibodies and the biotinylated secondary antibodies. (scale bars, 100 µm) *n* = 3 per group (**C**) The expression of GFAP, IBA-1, iNOS and COX-2 were detected by western blot using specific antibodies in the brain of mice. β-actin protein was used as an internal control and graphs represented the arbitrary density of blot signal. *n* = 4 per group. The data are shown as the means ± SD of the mean. ^#^
*p* < 0.05 control group vs. LPS group, * *p* < 0.05 LPS-group vs. LPS with AXT group.

**Figure 4 marinedrugs-17-00123-f004:**
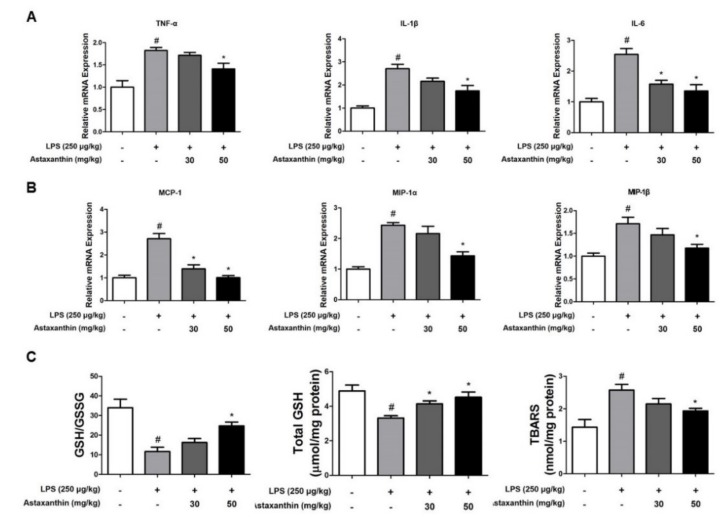
Effect of astaxanthin on LPS-induced pro-inflammatory cytokines, chemokines and oxidative stress in the brain of mice. mRNA expression levels of (**A**) pro-inflammatory cytokines such as TNF-α, IL-1β and IL-6 and chemokines (**B**) MCP-1, MIP-1α and MIP-1β in the brain of mice were measured using quantitative real-time RT-PCR (reverse transcription polymerase chain reaction). (**C**) GSH/GSSG ratio, total GSH and TBARS levels in the brain of mice were assessed using assay kit. *n* = 4 per group. The data are shown as the means ± SD of the mean. ^#^
*p* < 0.05 control group vs. LPS group, * *p* < 0.05 LPS-group vs. LPS with AXT group.

**Figure 5 marinedrugs-17-00123-f005:**
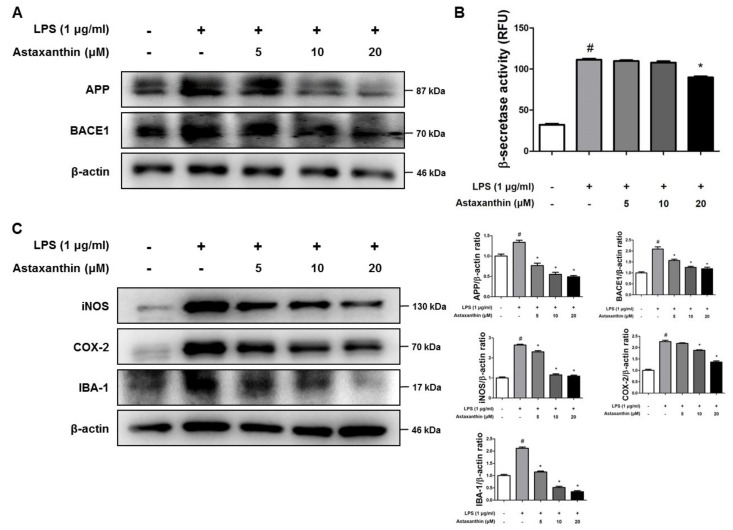
Effect of astaxanthin on amyloidogenesis and neuroinflammation in the microglia cell. Microglia BV-2 cells were treated with LPS (1 µg/mL) and AXT (5, 10 and 20 μM). (**A**) The expression of APP and BACE1 were detected by western blot using specific antibodies in the microglia BV-2 cells. β-actin protein was used as an internal control and graphs represented the arbitrary density of blot signal. (**B**) The levels of β-secretase activity in the microglia BV-2 cells were assessed using assay kit. (**C**) The expression of iNOS, COX-2 and IBA-1 were detected by western blot using specific antibodies in the microglia BV-2 cells. β-actin protein was used as an internal control and graphs represented the arbitrary density of blot signal. *n* = 3 per group; means ± SD of the mean. ^#^
*p* < 0.05 control group vs. LPS group, * *p* < 0.05 LPS-group vs. LPS with AXT group.

**Figure 6 marinedrugs-17-00123-f006:**
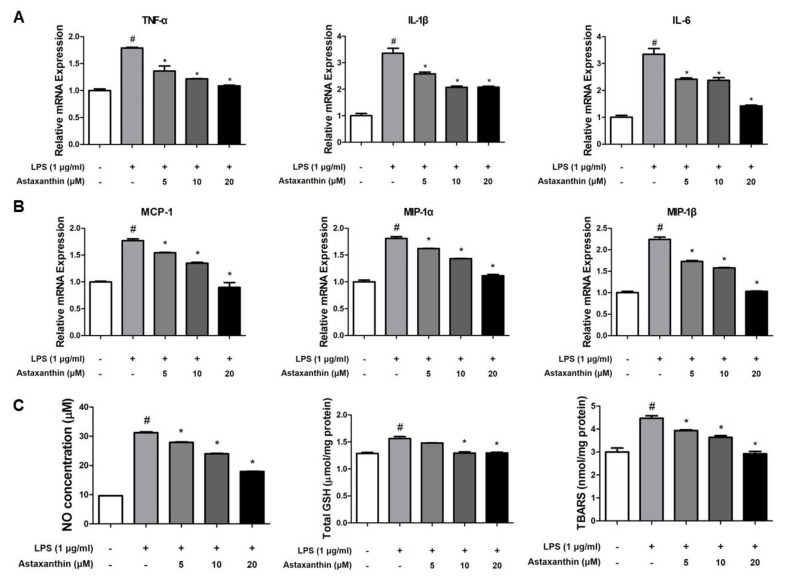
Effect of astaxanthin on LPS-induced pro-inflammatory cytokines, chemokines and oxidative stress in the microglia cell. mRNA expression levels of (**A**) pro-inflammatory cytokines such as TNF-α, IL-1β and IL-6 and chemokines (**B**) MCP-1, MIP-1α and MIP-1β in the microglia BV-2 cells were measured using quantitative real-time RT-PCR. (**C**) NO levels, total GSH and TBARS levels in the microglia BV-2 cells were assessed using assay kit. *n* = 3 per group; means ± SD of the mean. ^#^
*p* < 0.05 control group vs. LPS group, * *p* < 0.05 LPS-group vs. LPS with AXT group.

**Figure 7 marinedrugs-17-00123-f007:**
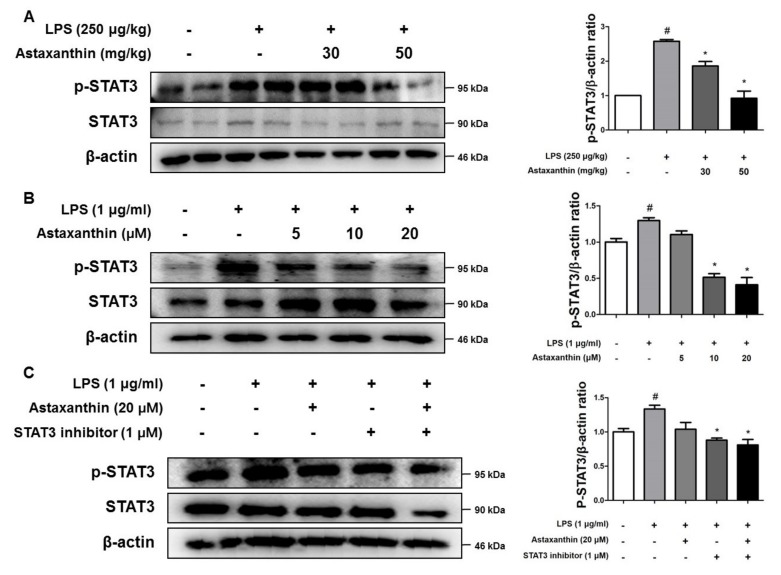
A vital role of STAT3 by binding astaxanthin LPS-induced neuroinflammation and oxidative stress in in vivo and in vitro. The expression of p-STAT3 and STAT3 were detected by western blot using specific antibodies in (**A**) the brain of mice (**B**) the microglia BV-2 cells. β-actin protein was used as an internal control and graphs represented the arbitrary density of blot signal. (**C**) The expression of p-STAT3 and STAT3 were detected by western blot using specific antibodies in the microglia BV-2 cells.

**Figure 8 marinedrugs-17-00123-f008:**
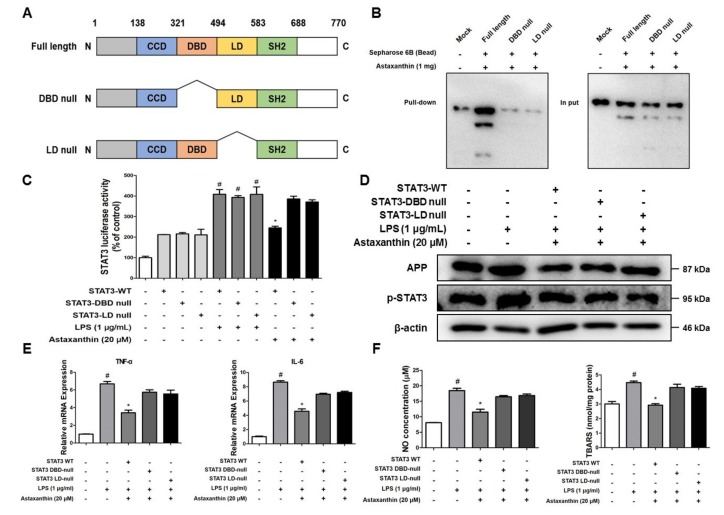
Astaxanthin directly binds to the DBD and LD domains of STAT3. (**A**) Schematic domain structures of STAT3 recombinant proteins. (**B**) Pull-down assay with deletion of binding sites of STAT3 was performed to determine whether AXT binds to DBD and LD domains of STAT3. (**C**) Luciferase assay was carried out to determine the effects of DBD null STAT3 and LD null STAT3 on the inhibitory effect of AXT on LPS-induced STAT3 luciferase activity. (**D**) The expression of p-STAT3 and STAT3 were detected by western blot using specific antibodies in the microglia BV-2 cells transfected with DBD null STAT3 and LD null STAT3. (**E**) mRNA expression levels of pro-inflammatory cytokines in the microglia BV-2 cells transfected with DBD null STAT3 and LD null STAT3 were measured using quantitative real-time RT-PCR. (**F**) NO concentration and TBARS levels in the microglia BV-2 cells transfected with DBD null STAT3 and LD null STAT3 were assessed using assay kit. *n* = 3 per group; means ± SD of the mean. ^#^
*p* < 0.05 control group vs. LPS group, * *p* < 0.05 LPS-group vs. LPS with AXT group.

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
