# Peer review of "Astaxanthin Ameliorates Lipopolysaccharide-Induced Neuroinflammation, Oxidative Stress and Memory Dysfunction through Inactivation of the Signal Transducer and Activator of Transcription 3 Pathway"

_marinedrugs, 2019, doi:10.3390/md17020123_

Reviewer 1 Report

In this manuscript, Han et al. reports the neuroprotective effect of astaxanthin against LPS –induced neuroinflammation, oxidative stress and memory dysfunction. Authors’ data indicate that astaxanthin ameliorates LPS-induced memory loss in a mouse model of exposure to LPS through inhibition of the activation of STAT3 pathway. This manuscript presents numerous data (illustrating a lot of experimental works with a total of 8 figures containing numerous panels) that could be interesting for scientific teams involved in this field of research. However, current data is not sufficient to support the conclusion of astaxanthin as an effective method of reducing the STAT3 pathway which might be a critical way in protecting brain against neuroinflammation, synaptic plasticity impairment and oxidative stress in the cortex and hippocampus. The authors need to modify their manuscript by addressing the following points.

Major points:

These is no data to indicate whether blood concentrations or brain tissue levels of astaxanthin in LPS-induced memory impairment mouse model have been altered after daily oral gavage for 4 weeks.

In Figure 7, it is not clear which phosphorylation antibody of STAT3 was used to detect p-STAT3 in the western blot. Please specify the phospho-STAT3 antibody or isoforms. Not fully understand why the level of phosphorylated STAT3 in Fig. 7A is much higher than the level of total STAT3.

In Figure 8, regarding the binding study, given the astaxanthin is a highly hydrophobic molecule, it is not clear that the astaxanthin is dissolved and chemically conjugated with Sepharose 4B beads.  

Minor points

In Introduction section, line 69, “The depletion of GSH attenuates the phosphorylation of STAT3 in cardiac myocytes”, is a confusion. In general, depletion of GSH would increase ROS production, please clarify this statement. In line 72, glutathione S-transferases should be used as GSH.

Author Response

Point 1: These is no data to indicate whether blood concentrations or brain tissue levels of astaxanthin in LPS-induced memory impairment mouse model have been altered after daily oral gavage for 4 weeks.

Response 1: Thank for comment. We performed high performance liquid chromatography analysis to detect concentration of brain tissue levels of astaxanthin. Brain tissue levels of astaxanthin is 68.191 ng/mg brain weight at 30 mg/kg and is 67.617 ng/mg brain weight at 50 mg/kg. However, there is no difference between 30 mg/kg administration group and 50 mg/kg administration group.

Point 2: In Figure 7, it is not clear which phosphorylation antibody of STAT3 was used to detect p-STAT3 in the western blot. Please specify the phospho-STAT3 antibody or isoforms. Not fully understand why the level of phosphorylated STAT3 in Fig. 7A is much higher than the level of total STAT3.

Response 2: Thank for comment. We used phospho-STAT3 antibody from Santa Cruz Biotechnology.

Point 3: In Figure 8, regarding the binding study, given the astaxanthin is a highly hydrophobic molecule, it is not clear that the astaxanthin is dissolved and chemically conjugated with Sepharose 4B beads.

Response 3: Thank for comment. We used Sepharose 6B beads and corrected Sepharose 4B to Sepharose 6B in material and method. Sepharose 6B can bind with hydrophobic molecule, thus it can be bind astaxanthin and Sepharose 6B.

Point 4: In Introduction section, line 69, “The depletion of GSH attenuates the phosphorylation of STAT3 in cardiac myocytes”, is a confusion. In general, depletion of GSH would increase ROS production, please clarify this statement. In line 72, glutathione S-transferases should be used as GSH.

Response 4: Thank for comment. We corrected sentences as followings; The depletion of GSH affected the phosphorylation of STAT3 in cardiac myocytes in line 69 and the blocking of STAT3 pathway protects against oxidative damages by increasing the level of GSH in line 72.

Reviewer 2 Report

This is an interesting study, however, this paper is not acceptable as it is.

There are some mistakes and insufficient expressions.  In addition, some of presented data are not clear.  More detailed expressions are needed.

Detailed some comments:

1. Results:

1-1  2.2. Astaxanthin downregulates……………:

 a) Line 136, “ BACE1 ” is appeared for the first time, thus the full name should be expressed.

 b) Figure 2C and line 138, The authors describe increase of APP by LPS injection was decreased by  AXT administration.  However, the result of 50 mg/kg of AXT seems to be increased not decreased (the left column of 50).  More clear results should be presented or describe the data variance.

1-2  2.3. Astaxanthin prevents…………..:

 Figure 3C and lines 158 - 160, In the same way mentioned above, the results of GFAP at 50 mg/kg of AXT seem to be increased compared to the results of 30 mg/kg of AXT.  More clear results should be presented or describe the data variance.

1-3  2.4. Astaxanthin reduces ………………:

 The effects of AXT on the decrease of GSH/GSSG and GSH by LPS injection are not explained; only the decrease of them by LPS is mentioned.  Description should be added.

1-4  2.5. Astaxanthin inhibits amyloidogenesis ….……………:

 Figure 5B and lines 204 - 205, The authors describe b-secretase activity was decreased by AXT dose-dependently.  However, it decreased only at 20 mM of AXT.  Description should be corrected.

1-5  2.6. Astaxanthin inhibits LPS-induced……………..:

 Figure 6C and line 228, The authors describe GSH levels were decreased by LPS, but increased by AXT co-treatment.  However, GSH increased by LPS in figure 6C.  Addition, it explain that GSH/GSSG ratio, total GSH and TBARS levels are shown in figure 6C in figure legends.  However, figure 6C shows NO, GSH and TBARS.  In line 227, the authors describe NO was increased by LPS.  Thus, figure legends are probably incorrect.  The authors should check and correct both figure 6 and sentences.  

1-6  2-7. Astaxanthin inhibits the phosphorylation…………..:

a)     Figure 7A and lines 241 - 242, The authors describe the LPS-induced activation of STAT3 was inhibited by AXT.  However, the inhibition by AXT is seen only at 50 mg/kg of AXT, STAT3 seems to be activated at 30 mg/kg of AXT.  More clear results should be presented or describe the data variance.

b)     Lines 243 - 245, The sentence “ These inhibitory ………….STAT3 inhibitor ” doesn’t’ t make sense.  In figure 7C, the activation of STAT3 by LPS is inhibited by AXT and also by STAT3 inhibitor, the inhibition by STAT3 inhibitor is more potent than AXT, and co-treatment with AXT and STAT3 inhibitor shows strong inhibition.  Thus, this sentence should be corrected.

1-7  2.8. Astaxanthin directly………………:

 a)  “ DBD ” and “ LD ” are appeared for the first time, thus their full name should be expressed.

b)  The method or analysis soft for virtual docking analysis should be specified.

c)      Which company was the epoxy-activated Sepharose 6B beads obtained from?  Also, the procedure to produce AXT-conjugated beads should be described.

d)     I think the index of binding affinity such as “ kcal/mol ” is not familiar.  Thus, explanation of the meaning of “ -9.0 kcal/mol ” is needed to understand this section.  Show an estimated value of KD , if it is possible. 

e)     Figure 8D and lines 272 – 274, The authors describe the expression levels of APP and p-STAT3 was not decreased in the cells transfected with DBD-null and LD-null STAT3 by AXT.  However, those levels seem to be decreased compared with the levels of LPS treatment alone (the second column from left) and the levels are almost same as wild-type STAT3 (the middle column) in figure 8D.  More clear results should be presented or describe the data variance.

2. Discussion:

 Lines 336 – 337, The sentence “ In the present study, ……………….was decreased “ doesn’t make sense.  The recover of the GSH/GSSG ratio and GSH level are due to AXT treatment.  Thus, this sentence should be revised.

Author Response

Point 1: 2.2. Astaxanthin downregulates……………:

a)     Line 136, “BACE1” is appeared for the first time, thus the full name should be expressed.

b)    Figure 2C and line 138, The authors describe increase of APP by LPS injection was decreased by AXT administration. However, the result of 50 mg/kg of AXT seems to be increased not decreased (the left column of 50). More clear results should be presented or describe the data variance.

Response 1: Thank for comment. We corrected sentences as following; We investigated the level of APP and β-secretase (BACE1) proteins using western blot analysis in line 136 and the expression of APP was decreased in the 30 mg/kg AXT administration group and the expression of BACE1 was reduced by the administration of AXT in line 138.

Point 2: 2.3. Astaxanthin prevents……………:

Figure 3C and lines 158 – 160, In the same way mentioned above, the results of GFAP at 50 mg/kg of AXT seem to be increased compared to the results of 30 mg/kg of AXT. More clear results should be presented or describe the data variance.

Response 2: Thank for comment. We wrote the sentence as following; However, the expression of GFAP was decreased at 30 mg/kg in the AXT-administered mice (Figure 3C).

Point 3: 2.4. Astaxanthin reduces……………:

The effects of AXT on the decrease of GSH/GSSG and GSH by LPS injection are not explained; only the decrease of them by LPS is mentioned. Description should be added.

Response 3: Thank for comment. We wrote the effects of AXT as following; but it was increased by AXT treatment.

Point 4: 2.5. Astaxanthin inhibits amyloidogenesis……………:

Figure 5B and lines 204 – 205, The authors describe β-secretase activity was decreased by AXT dose-dependently. However, it decreased only at 20 mM of AXT. Description should be corrected.

Response 4: Thank for comment. We revised description as following; The LPS-induced β-secretase activity was decreased in the AXT-treated BV-2 cells at 20 μM.

Point 5: 2.6. Astaxanthin inhibits LPS-induced……………:

Figure 6C and line 228, The authors describe GSH levels were decreased by LPS, but increased by AXT co-treatment.  However, GSH increased by LPS in figure 6C.  Addition, it explain that GSH/GSSG ratio, total GSH and TBARS levels are shown in figure 6C in figure legends.  However, figure 6C shows NO, GSH and TBARS.  In line 227, the authors describe NO was increased by LPS.  Thus, figure legends are probably incorrect.  The authors should check and correct both figure 6 and sentences.

Response 5: Thank for comment. We revised the sentences as following; The total GSH level was increased by LPS and was decreased by AXT treatment. However, there is no big difference between each group (Figure 6C). We also corrected the figure legends 6 as following; NO levels, total GSH and TBARS levels in the microglia BV-2 cells were assessed using assay kit.

Point 6: 2.7. Astaxanthin inhibits the phosphorylation……………:

a)     Figure 7A and lines 241 - 242, The authors describe the LPS-induced activation of STAT3 was inhibited by AXT.  However, the inhibition by AXT is seen only at 50 mg/kg of AXT, STAT3 seems to be activated at 30 mg/kg of AXT.  More clear results should be presented or describe the data variance.

b)    Lines 243 - 245, The sentence “ These inhibitory …………. STAT3 inhibitor ” doesn’t’ t make sense.  In figure 7C, the activation of STAT3 by LPS is inhibited by AXT and also by STAT3 inhibitor, the inhibition by STAT3 inhibitor is more potent than AXT, and co-treatment with AXT and STAT3 inhibitor shows strong inhibition.  Thus, this sentence should be corrected.

Response 6: Thank for comment. We corrected the sentences as following; In the presence of AXT, the LPS-induced phosphorylation of STAT3 was inhibited in the LPS-injected mice at 50 mg/kg (Figure 7A) in line 241 – 242 and the inhibition of phosphorylated STAT3 by STAT3 inhibitor was more effective than by AXT (Figure 7C) in line 243 – 245.

Point 7: 2.8. Astaxanthin directly……………:

a)      “ DBD ” and “ LD ” are appeared for the first time, thus their full name should be expressed.

b)    The method or analysis soft for virtual docking analysis should be specified.

c)     Which company was the epoxy-activated Sepharose 6B beads obtained from?  Also, the procedure to produce AXT-conjugated beads should be described.

d)     I think the index of binding affinity such as “ kcal/mol ” is not familiar.  Thus, explanation of the meaning of “ -9.0 kcal/mol ” is needed to understand this section.  Show an estimated value of K, if it is possible.

e)     Figure 8D and lines 272 – 274, The authors describe the expression levels of APP and p-STAT3 was not decreased in the cells transfected with DBD-null and LD-null STAT3 by AXT.  However, those levels seem to be decreased compared with the levels of LPS treatment alone (the second column from left) and the levels are almost same as wild-type STAT3 (the middle column) in figure 8D.  More clear results should be presented or describe the data variance.

Response 7: Thank for comment. We wrote the full name of DBD and LD in the manuscript and marked. We showed the method of virtual docking analysis in the manuscript and marked. We also wrote company of the epoxy-activated Sepharose 6B in the material and method and marked. We erased the sentence “however, the expression levels of these proteins were not decreased in the BV-2 cells transfected with DBD-null STAT3 and LD-null STAT3 vector and treated with AXT. And we presented this sentence “however, the expression levels of these protein were more decreased in the BV-2 cells transfected with wild-type STAT3 vector than in the BV-2 cells transfected with DBD-null STAT3 and LD-null STAT3 vectors and treated with AXT. It looks same as between wild-type STAT3 and DBD-null STAT3, LD-null STA3, but the expression of APP and p-STAT3 in wild-type STAT3 group was smaller than in DBD-null STAT3 or LD-null STAT3 group compared to β-actin. Because the expression of β-actin in wild-type STAT3 group was larger than in DBD-null STAT3 or LD-null STAT3 group.

Point 8: Discussion: Lines 336 – 337, The sentence “ In the present study, ……………….was decreased “ doesn’t make sense.  The recover of the GSH/GSSG ratio and GSH level are due to AXT treatment.  Thus, this sentence should be revised.

Response 8: Thank for comment. We revised the sentence as following; In the present study, the GSH/GSSG ratio and the total GSH level were observed to recover, and the TBARS level was decreased by AXT administration.
